# Half Friend, Half Enemy? Comparative Phytophagy between Two Dicyphini Species (Hemiptera: Miridae)

**DOI:** 10.3390/insects13020175

**Published:** 2022-02-07

**Authors:** Paula Souto, Gonçalo Abraços-Duarte, Elsa Borges da Silva, Elisabete Figueiredo

**Affiliations:** 1Instituto Superior de Agronomia (ISA), Universidade de Lisboa, Tapada da Ajuda, 1349-017 Lisboa, Portugal; gduarte@isa.ulisboa.pt (G.A.-D.); elsasilva@isa.ulisboa.pt (E.B.d.S.); 2Linking Landscape, Environment, Agriculture and Food (LEAF), Instituto Superior de Agronomia, Universidade de Lisboa, Tapada da Ajuda, 1349-017 Lisboa, Portugal; 3Forest Research Centre (CEF), Instituto Superior de Agronomia (ISA), Universidade de Lisboa, Tapada da Ajuda, 1349-017 Lisboa, Portugal

**Keywords:** omnivorous predator, *Nesidiocoris tenuis*, *Dicyphus cerastii*, plant damage, zoophytophagy, fruit injury, protected crops, biological control, tomato

## Abstract

**Simple Summary:**

Omnivorous predators, such as some mirids, are important biological control agents in several vegetable crops since they are generalists and can survive in the crop in the absence of prey. *Nesidiocoris tenuis* is a mirid used worldwide and its phytophagy is well known, which is not the case for the Palearctic *Dicyphus cerastii.* To use the latter in biological control it is crucial to evaluate the damage it causes to plants. We compared these two mirid species, under laboratory and semi-field conditions, regarding the damage they caused to plants and fruit, and their location on the plant versus on the fruit. Both species produced plant damage (scar punctures on leaves and necrotic patches on petioles) and caused flower abortion, at a similar level, however, only *N. tenuis* produced necrotic rings. Overall, *N. tenuis* females produced more damage to tomato fruit than *D. cerastii*. There was an increased frequency of *D. cerastii* females found on the plants over time, which did not happen with *N. tenuis*. Our results suggested that although *D. cerastii* caused less damage to tomato fruit than *N. tenuis*, it did feed on the fruit and could cause floral abortion, which requires field evaluation and caution in its use.

**Abstract:**

Despite their importance as biological control agents, zoophytophagous dicyphine mirids can produce economically important damage. We evaluated the phytophagy and potential impact on tomato plants of *Dicyphus cerastii* and *Nesidiocoris tenuis*. We developed a study in three parts: (i) a semi-field trial to characterize the type of plant damage produced by these species on caged tomato plants; (ii) a laboratory experiment to assess the effect of fruit ripeness, mirid age, and prey availability on feeding injuries on fruit; and (iii) a laboratory assay to compare the position of both species on either fruit or plants, over time. Both species produced plant damage, however, although both species produced scar punctures on leaves and necrotic patches on petioles, only *N. tenuis* produced necrotic rings. Both species caused flower abortion at a similar level. Overall, *N. tenuis* females produced more damage to tomato fruit than *D. cerastii*. There was an increased frequency of *D. cerastii* females found on the plants over time, which did not happen with *N. tenuis*. Our results suggested that, although *D. cerastii* caused less damage to fruit than *N. tenuis*, it still fed on them and could cause floral abortion, which requires field evaluation and caution in its use in biological control strategies.

## 1. Introduction

Zoophytophagous mirid species (Hemiptera: Miridae) are important biological control agents in several crops. Dicyphine (Miridae: Bryocorinae: Dicyphini) species, such as *Nesidiocoris tenuis* (Reuter), and several species of the genera *Macrolophus* Fieber and *Dicyphus* Fieber, are used worldwide as generalist predators [1] on several vegetable crops, both in conservation and augmentative biological control strategies.

*Dicyphus cerastii* Wagner is a Palearctic mirid, reported in the Mediterranean Basin, which spontaneously colonizes Portuguese greenhouses [2]. It is currently being evaluated as a candidate biological control agent (BCA), since it can feed on several horticultural pests [3]. *Nesidiocoris tenuis* is currently commercialized and released to control whiteflies and *Tuta absoluta* (Meyrick) in Mediterranean greenhouses [1,4,5]

Dicyphine mirids may resort to phytophagy in periods of prey scarcity [6,7], and to obtain water [8] and nutrients [9] from plants. Despite being advantageous as a feeding strategy, phytophagy can have negative effects in an agronomical context. Plant feeding may lead to a decrease in predation activity [9,10], and, more importantly, phytophagy can cause damage of economic importance, such as necrotic rings in stems and leaf petioles, as well as flower or fruit abortion, and punctures in the fruit [6,11,12,13,14]. This is particularly evident with *N. tenuis*, which is often the target of pesticide sprays that are used to control its populations when there is a risk of plant damage occurring, a practice that negatively impacts other natural enemies present on crops.

The increasing demand for food products without pesticide residues, combined with the need to control pests, highlights the urgency for sustainable alternatives that reduce negative effects on both the consumer and the environment’s health. The use of predatory mirids in biological control has been very successful in protected tomato crops (e.g., [5,15,16]), despite the damage produced by some species. Therefore, to enhance biological control in tomato crops, research should focus on less phytophagous yet efficient dicyphine predators.

Plant damage can greatly vary with host plant species [6], mirid species [6,17,18], and even among populations of the same species [19,20]. Therefore, understanding the risk that phytophagy represents to crops is a key point in the evaluation of a new candidate BCA [13], such as *D. cerastii*, that will help decision making while choosing the best suited mirid species. Although damage caused by *N. tenuis* has been studied (e.g., [6,11,21,22]), little is known about those induced by *D. cerastii.* The latter has been observed producing chlorotic punctures on leaves [4] and also necrotic damage on tomato stems and leaf petioles, as well as feeding punctures on fruit (our pers. obs.). However, the influence of its damage on plant development and possible economic impacts has never been evaluated or compared with other mirid species.

We hypothesized that *D. cerastii* and *N. tenuis* display different phytophagous behavior. Within this context, the aim of this study was to compare the phytophagy and potential impact on tomato production of these two Dicyphini species. For this, we characterized the type of plant damage produced by these species on tomato plants. Then, we assessed the effect of fruit ripeness, mirid age, and prey availability on feeding injuries on tomato fruit. Finally, we compared the location of both species on either tomato fruit or plants.

## 2. Materials and Methods

### 2.1. Rearing of Mirid Predators

Stock colonies of both species (*D. cerastii* and *N. tenuis*) are maintained at the Instituto Superior de Agronomia (ISA), Lisbon, Portugal, on tobacco plants (*Nicotiana tabacum* L.). The colony of *D. cerastii* was started with individuals from different geographical sites in Portugal (Fataca, Ferreira do Zêzere, Lisbon, and Póvoa de Varzim) and is frequently refreshed with individuals, mostly from the Oeste region (Mafra and Silveira). The colony of *N. tenuis* was started with individuals from the Oeste region (Silveira) and from Koppert Biological Systems (The Netherlands). For rearing details, see [3]. Young adult females (between 1 and 7 days old), for all three bioassays, were obtained from the regular collection of large nymphs from breeding cages that were transferred to separate cages, where they could reach adulthood. For nymph experiments (see Section 2.3, Fruit Damage), 4th/5th instar nymphs were collected from immature rearing cages.

### 2.2. Phytophagy in Semi-Field Conditions

Phytophagy was observed for *N. tenuis* and *D. cerastii* in semi-field conditions in a greenhouse at ISA’s *campus*. For this, mesh cages (1.5 m high and 1.0 m wide) were used. In each cage, there were two tomato plants (cv. Vayana), each one in a 15 L pot. Plants were fertilized and watered as needed, using an organic fertilizer solution (Húmus Líquido Horta^®^, SIRO, Mira, Portugal). When plants had 4 to 5 developed leaves (ca. 30–40 cm high), six couples of *D. cerastii* or *N. tenuis* were released. Control cages were set without any insects. Each treatment (12 *D. cerastii*, 12 *N. tenuis* and a control) had five replicates. To simulate the natural presence of prey, a teaspoon (ca. 4 g) of a mix of *Artemia* spp. (Anostraca: Artemiidae) and *Ephestia kuehniella* Zeller (Lepidoptera: Pyralidae) eggs (Entofood^®^, Koppert Biological Systems) was sprinkled evenly on the plants at the time of insect release, and every two weeks after that. Six weeks after release, all insects in the cages were captured into vials containing 96% ethanol and counted. Plant damage, expressed as the total number of necrotic rings or necrotic patches, was recorded for every cage. Flower abortion was counted as the proportion of missing flowers/total number of flowers in the flower rachis, for every cage. When trusses already had small fruit, these were counted as flowers. The mirids were released on 28 April 2021, and the assay ended on 11 June 2021.

### 2.3. Fruit Damage

The following factors were considered to compare the puncture level on fruit: mirid species (*D. cerastii* and *N. tenuis*), mirid age (nymph and adult female), tomato ripeness (unripe and fully ripe), and availability of food and/or water. For each species, three adult females or three nymphs were placed in plastic cups (8 cm high and 6 cm diameter) with one tomato fruit that was approximately 4–5 cm long. The lid of the cups had a hole (3 cm ⌀) covered with fabric to allow ventilation. Four treatments were considered (15 replications), for each species, mirid age, and tomato ripeness: (a) fruit only (N); (b) fruit with water (W) (supplied through an Eppendorf vial with moist cotton wool); (c) fruit with water and alternative food (FW) (*Ephestia kuehniella* eggs and *Artemia* spp. cysts Entofood^®^, Koppert Biological Systems, Berkel en Rodenrijs, The Netherlands), supplied on a sticky paper strip (2.0 cm × 0.8 cm); (d) fruit with alternative food (F) (Entofood^®^) but no water. Tomato fruit (Figure 1, cv. Suntasty) were obtained from an organic commercial tomato greenhouse. Before the experiment, the fruit were washed with abundant water and individually inspected for any marks or possible feeding punctures. Fruit with any defect were discarded. The unripe condition was considered as fully grown green fruit and fully ripe was considered as fully red fruit. For each bioassay, a control (i.e., only fruit) was made to ensure that punctures were caused by mirids. Insects were allowed to feed for 24 h, after which punctures (injuries resulting from fruit feeding) were counted under a stereomicroscope with a magnification of 50×. Injury was considered as a puncture surrounded by a small whitish or yellowish halo [12,23]. Replicates in which death occurred, or nymphs molted into adults, were discarded.

### 2.4. Location on Tomato Plant versus Fruit

The tomato fruit were the same cultivar as those in the fruit-damage bioassay (see Section 2.3, above). Females of both species were placed individually in 600 mL transparent plastic cups covered with fabric to allow ventilation. The cups contained a young tomato plant (cv. San Pedro) held in water in a 10 mL glass bottle, and an unripe (green) tomato fruit. The unripe tomato was elected after analyzing the data from the tomato-damage bioassay, in which the most injured tomato was unripe. Females were placed individually for 24 h before the experiment in empty test tubes (starved). After release into the plastic cups, the position of the females was recorded at 1 h, 2 h, 6 h, and 24 h as being on the young plant, on the fruit, or elsewhere in the cup (as proposed by McGregor et al. [24]). Observations were conducted in a controlled chamber (Fitoclima CP500, Aralab Lda., Albarraque, Sintra, Portugal) at a temperature of 25 °C and photoperiod of 14 h. At the end, fruit were inspected for feeding punctures. Females were used only once. In total, 20 replicates were made for each species.

### 2.5. Data Analysis

Phytophagy in semi-field conditions. Both *D. cerastii* and *N. tenuis* populations at the end of the experiment and plant damage (necrotic rings or necrotic patches) were compared between species using one-way ANOVA with species as an independent variable. Differences in flower abortion among treatments were compared with Pearson’s χ^2^ tests. These statistical analyses were performed with IBM SPSS statistics v.26 (IBM, Armonk, NY, USA).

Fruit damage bioassay. Classification tree methods were used to understand the relative importance of the variables used (i.e., the absence or presence of water and/or food, developmental stage *of* the mirid, tomato ripeness, and species) on the number of feeding punctures. Statistical analyses were performed using R software version 4.1.0 implemented in RStudio version 1.4.1106. Conditional inference trees were made using the “*ctree*” function (R package party, http://cran.r-project.org/web/packages/party/index.html, accessed on 9 June 2021), which bases node splitting on statistical tests, providing a *p*-value for the significance of splitting [25]. The importance of the variables was measured using the random forest algorithm [26] and computations were performed in the randomForest package with 1001 trees (ntree = 1001). The random forest algorithm combines many classification trees to produce more accurate classifications and has measures of variable importance and measures of similarity of data points as by-products of its calculations [27]. Data were analysed together (i.e., considering counts for both species) and separately for each species. All preliminary analysis considered the modalities in two groups: (i) without food (N and W); and (ii) with food (F and FW). We grouped them into absence (A) and presence (P) of food and/or water, respectively, in the presented output.

Position on tomato plant versus fruit. The position of insects was compared between species for each observation time using the Fisher’s exact test and z-test with Bonferroni correction method. For the comparison among locations within each species and each observation time, the non-parametric χ^2^ test for one sample was used. These statistical analyses were performed with IBM SPSS statistics v.26 (IBM, Armonk, NY, USA).

## 3. Results

### 3.1. Phytophagy in Semi-Field Conditions 

The average temperature during the assay was 24.3 °C, with a minimum of 10.4 °C and a maximum of 45.4 °C. The relative humidity was ca. 50%. At the end of the experimental period, *N. tenuis* had a larger average population (137.4 ± 30.3 individuals/cage) than *D. cerastii* (68.4 ± 14.4 individuals/cage); this difference, however, was not significant (F = 4.182; df = 1; *p* = 0.075).

*Nesidiocoris tenuis* produced both necrotic patches (ca. 10% of total damage) on leaves and stems and necrotic rings (ca. 90% of total damage) (Figure 1), whereas *D. cerastii* only produced necrotic patches (Figure 2). Necrotic rings caused by *N. tenuis* occasionally led to withering of young shoots or leaves (Figure 1b), while this was not observed with *D. cerastii* patches. Plant damage numbers were significantly different between species (F = 17.114; df = 1; *p* = 0.003), and *N. tenuis* produced more necrotic injuries on the plants than *D. cerastii* (Figure 3).

Both species also produced puncture scars on leaves as a result of feeding on young stem/leaf tissues (Figure 2a).

Control cages had significantly lower flower abortion than both *D. cerastii* (χ^2^ = 12.047; df = 1; *p* = 0.001) and *N. tenuis* cages (χ^2^ = 16.395; df = 1; *p* < 0.001), whereas the two mirid species displayed similar levels of flower abortion (χ^2^ = 0.670; df = 1; *p* = 0.413) (Figure 4).

### 3.2. Fruit Damage

Both species fed on tomato fruit and produced punctures that appeared as damaged epicarp/mesocarp cells. It was often possible to observe that the damaged area extended beyond the puncture point following stylet movement inside the fruit. Punctures were structurally similar (Figure 5), but the pattern of each species was different. *Dicyphus cerastii* punctures tended to be aggregated, forming clearly visible patches in cases of highly damaged fruit. *Nesidiocoris tenuis* punctures appeared less aggregated compared to *D. cerastii*. Punctures on fruit did not heal, as punctures on green fruit did not disappear even when fruit changed color during maturation. Punctures produced by females and nymphs appeared similar, for both species. We observed that, occasionally, females of both species laid eggs on fruit. It is possible that an amount of the punctures may have been egg laying attempts or the result of probing.

A high variability was observed in the number of punctures inflicted in all treatments and in both species, with low numbers or even no punctures to high numbers of punctures in the same treatment, which generated great variance in the data for both species studied. When analyzing the dataset considering both species or for each species separately, there was no difference between the two treatments without food (N and W) or between the two treatments with food (F and FW). Considering both species, the most important variable was the tomato ripening stage (tomato_age), with the unripe (green) tomato being the one with the highest number of feeding punctures, followed by the presence/absence of food. Food presence (F, FW) or absence (N, W) was only significant in the case of green fruit, and species was only significant for females in the presence of food in the case of green fruit, and for females in the ripe fruit. While the most important factor for *D. cerastii* was also the tomato ripeness, for *N. tenuis* the most important factor was the individual’s stage of development (i.e., whether it is a nymph or an adult). Food was the second factor for *D. cerastii* but only the third, and more distant, for *N. tenuis* (Figure 6 and Figure 7).

### 3.3. Location on Tomato Plant vs. Fruit

The locations of *N. tenuis* and *D. cerastii* were only significantly different at the first observation (1 h) (Fisher’s exact test value = 6.423, *p* = 0.033), with the former more present on the young plant than the latter and the inverse regarding the cup (α = 0.05) (Figure 8). Both *N. tenuis* and *D. cerastii* were mainly found on the young plant. However, in the case of *D. cerastii*, differences among locations were only verified at 2 h, 6 h, and 24 h, with females more present on the young plant (or on the young plant or on the cup walls) than on the fruit. In the case of *N. tenuis*, the females were more often found on the young plant than on fruit or cup walls, except at 24 h; at this time there were no differences between the young plant and fruit and no female was observed on cup walls.

In this bioassay, damage on plants was not quantified since, in some treatments, they would not be identified, especially with *D. cerastii*. Feeding punctures on fruit were found (although they were not counted) in all cases when the females of both species were sighted on the fruit. Furthermore, in both species, at least one fruit with feeding punctures was found, although the female was never observed in that same fruit during the bioassay (one case in *N. tenuis* and two cases in *D. cerastii*).

## 4. Discussion

The occurrence of plant damage production by zoophytophagous mirids is influenced by factors such as prey scarcity (e.g., [11,21,28]), water stress (e.g., [8]), and also host plant and mirid species (e.g., [6]).

Phytophagous behaviour in *N. tenuis* is regarded as more severe than in other dicyphine mirids [6,17,18] since this species has the particularity to feed on vascular tissues and to aggregate on feeding sites [29]. Our study corroborates this, as plants with *N. tenuis* suffered more damage than those with *D. cerastii*. Moreover, the higher number of necrotic rings produced by *N. tenuis* were also more severe for the plant compared to the necrotic patches observed for *D. cerastii*. We observed that, in some *N. tenuis* infested plants, the apical shoots or leaflets withered because of necrotic rings in stems, whereas this was not observed in *D. cerastii* plants.

Besides the damage to vegetative parts of the plant, dicyphines are also reported to damage reproductive organs, such as flowers and fruit [6]. *N. tenuis* is also recognized for causing flower and fruit abortion on tomato plants [30], and in a study by Sanchez et al. [31] the percentage of flower abortion by *N. tenuis* reached up to 50% during population peaks. Flower and fruit abortion on tomato plants has also been reported for *M. pygmaeus* [32]. However, to our knowledge, this type of damage has not been previously described for *Dicyphus* spp. In our experimental conditions, we found that the percentage of flower abortion was not different between both mirid species. As flower abortion is particularly important on cluster tomato cultivars, *D. cerastii* may have a similar impact to *N. tenuis* on such cultivars, despite the lower damage to vegetative tissues produced by *D. cerastii*.

In this study, the presence of water did not influence fruit puncture level. Therefore, both mirid species and both development stages could obtain the water they needed from green or ripe tomato fruit, at least when water was not provided. Water provision has been reported as one reason for phytophagy on heteropteran predators (e.g., [9]). However, as puncture numbers did not differ when water was supplied for both *N. tenuis* and *D. cerastii*, these mirids may have looked for other resources when they fed on the fruit.

Among the nutrients obtained from phytophagy, carbohydrates may have a particular ecological function since they have been reported to influence both predation and reproduction in dicyphines. This was demonstrated for *N. tenuis*, which was able to reduce the amount of prey feeding needed to establish itself on tomato plants [5], and increased its progeny [33], in the presence of sucrose dispensers. In another study, *N. tenuis* reduced its phytophagy when provisioned with sucrose dispensers [34].

In our study, when considering both species combined, tomato ripeness was the most important factor, with green fruit suffering more punctures than mature ones. This difference may be due to distinct nutritional profiles between unripe and ripe fruit. Sugar concentration, among other nutrients, may be higher in ripe tomato fruit [35,36], so it is possible that mirids may obtain more nutritional value per feeding puncture on ripe fruit than on green ones. On green fruit, the main effect was the presence of prey, which reduced fruit damage. There were differences between the species as *N. tenuis* females produced more damage than those of *D. cerastii*. On ripe fruit the most important factor was mirid age, with females producing most damage and, in these fruit, food did not significantly reduce damage. However, and once again, females of *N. tenuis* produced more damage than those of *D. cerastii*.

Considering *N. tenuis*, the most important factor on fruit damage was age, with females damaging more fruit than nymphs. Differently, in studies with whole plants, *N. tenuis* nymphs showed higher carbohydrate content [34] and spent more time feeding on the apical part of the plant [20], compared to adults. A similar trend was found for nymphs of the neotropical mirids, such as *Macrolophus basicornis* (Stål), *Engytatus varians* (Distant) and *Campyloneuropsis infumatus* (Carvalho), that also produced fruit punctures, whereas females did not [23]. Even though we observed less fruit damage by *N. tenuis* nymphs than females, our results indicated that nymphs were less influenced by factors such as fruit ripeness or presence of prey, suggesting that *N. tenuis* nymphs may be less prone to change their phytophagous behavior than adults. Following mirid age, fruit ripeness was the next important factor for *N. tenuis*, with green fruit sustaining more damage. The presence of prey reduced the amount of damage on green fruit, whereas on ripe fruit it did not produce differences. This further suggests that green fruit may be a less valuable nutritional source for *N. tenuis*.

Tomato ripeness was the most relevant factor to explain fruit punctures by *D. cerastii*. The presence of prey was also important for fruit damage reduction in green fruit. Differently to *N. tenuis,* mirid age was not important in this species, which may suggest that *D. cerastii* may not be as dissimilar in phytophagy between adults and nymphs as *N. tenuis*.

Plant damage by zoophytophagous mirids has been associated with prey scarcity [11,21]. However, in our study the presence of food did not affect puncture level on ripe or green fruit with *N. tenuis* nymphs. Similarly, McGregor et al. [24] reported that the presence of food did not influence the level of fruit feeding by *Dicyphus hesperus* on mature tomato fruit, and Lucas and Alomar [37] reported that in whole caged plants the presence of *E. kuehniella* eggs did not prevent fruit injury by *D. tamaninii*.

Plant damage by zoophytophagous mirids may be determined by a complex combination of factors, besides prey abundance. Different species may have distinct preference or behavior that produce different types and levels of damage. Under the same conditions *M. caliginosus* (in fact, *M. pygmaeus*, C. Castañé, pers. comm.) did not produce fruit damage, whereas *D. tamaninii* did [37]. A different dicyphine, the nearctic *D. hesperus* preferred to feed on tomato leaves producing negligible damage on fruit [24]. Host plant and cultivar may also determine phytophagy, as was demonstrated for *N. tenuis*, which varied its phytophagy among different tomato cultivars [38]. The health of the host plant may also shape phytophagy by dicyphines. *Macrolophus pygmaeus* was reported to increase the number and produce more evident fruit damage on tomato plants infected with Pepino mosaic virus (PepMV) [12], but the same did not occur with *N. tenuis* [14]. Defence-activated plants may also be less susceptible to mirid phytophagous behavior. This was demonstrated for *N. tenuis*, which produced less plant damage on tomato plants inoculated with the endophytic *Fusarium solani* K strain, a fungal isolate that confers tomato resistance to foliar and root fungal pathogens [39].

Other factors may explain differences in phytophagy, such as genetic variation within species [19,20]. In fact, for the same treatments, we observed high variability in puncture numbers inflicted on fruit. As the large majority of the individuals used to initiate, and all the ones used to refresh the rearings, came from nearby locations (less than 45 km of linear distance), it is likely that the geographic origin was not a key determining factor in the high variability in feeding puncture number. This suggests that other factors, other than those considered in our study, may be driving fruit feeding in both *N. tenuis* and *D. cerastii*, and genetically determined behaviors should probably be considered in future research.

The fact that most *N. tenuis* and *D. cerastii* females were found on the young plants rather than elsewhere in the cup, and that there was an increased frequency of *D. cerastii* females found on young plant over time, may be related to the search for a better oviposition site [24]. Despite this, we could observe a slight increase over time of *N. tenuis* females occurring on fruit, which became the same as that for young plants at 24 h, suggesting a potential risk to fruit by this species. Furthermore, although few females of both species were observed on fruit compared to plant parts, feeding punctures were observed on fruit where females were not seen throughout the observations, showing that the female was at some moment on the fruit and fed on it. Finally, as *D. cerastii* preferred to be on young tomato plants than tomato fruit for plant feeding over time, the potential for damage to tomato fruit by this zoophytophagous mirid may be lower when compared to *N. tenuis.*

In the field, in commercial, protected tomato crops, necrotic rings, shoot, and flower cluster withering, and also punctures on fruit, are common when *N. tenuis* is present at high densities. This has repercussions on tomato production, leading growers to use a tolerance threshold and apply control measures. In the case of *D. cerastii,* necrotic tissues and punctures on fruit have been observed in the field by our team in commercial greenhouses when this species is present in high population densities. In order to fully assess how *D. cerastii* may affect tomato production (both in quantity and quality), further research is needed in semi-field conditions and commercial greenhouses, to establish safe population density thresholds. Since the damage caused by *D. cerastii* was apparently different from the necrotic rings of *N. tenuis,* histological studies are needed to characterize the necrotic patch damage reported here. Furthermore, it is important to understand if the feeding behaviour of *D. cerastii* induced the production of volatile defence compounds in the damaged plant, as reported for *N. tenuis*, *M. pygmaeus* and *Dicyphus maroccanus*, Wagner (syn. *D. bolivari* Lindberg) [40,41], with the consequent attraction of other biological control agents [41]. 

## 5. Conclusions

Overall *D. cerastii* damage was less severe than *N. tenuis*, as it did not cause necrotic rings and was more likely to seek out parts of the plant than the fruit. Despite this, it fed on fruit and caused flower abortion. Therefore, as was already known for *N. tenuis* and *M pygmaeus*, *D. cerastii* has the potential to cause an economic impact on tomato fruit production, particularly for cluster tomato cultivars, since its damage is related to the parts of the plant responsible for fruit production. We suggest that decision making regarding its use as a biological control agent should be made through field evaluation considering different cultivars. 

We also found that fruit damage was highly variable within treatments, indicating that there may be differences in phytophagy on both species and individual levels. Therefore, in the future, selection of less phytophagous populations/strains combined with adequate management strategies may also benefit from the predatory behavior of dicyphine mirids with lower negative impact on tomato production.

## Figures and Tables

**Figure 1 insects-13-00175-f001:**
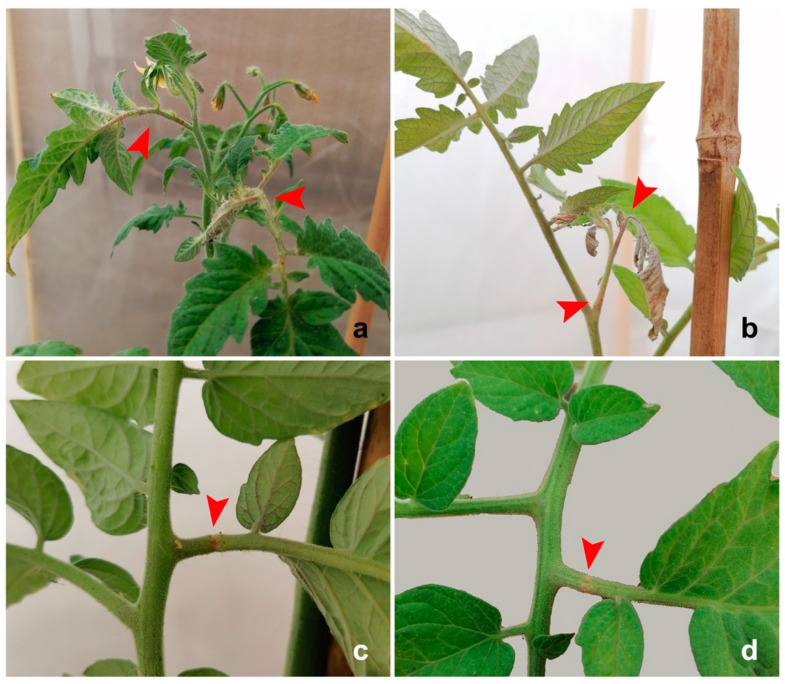
Plant damage by *Nesidiocoris tenuis* on tomato plants: (**a**) necrotic rings on leaves and shoots; (**b**) shoot wilting; (**c**,**d**) detail of necrotic rings on leaves.

**Figure 2 insects-13-00175-f002:**
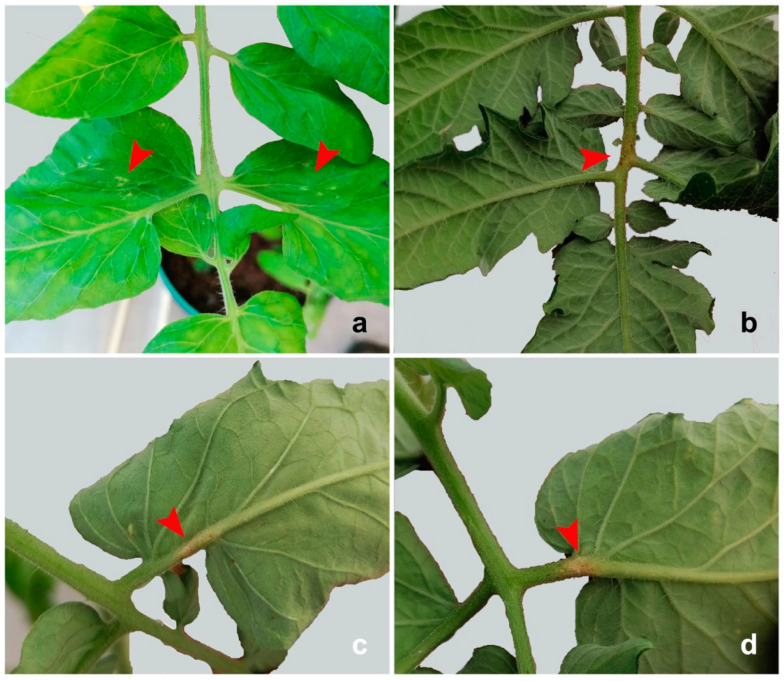
Plant damage by *Dicyphus cerastii* on tomato plants: (**a**) puncture scars on expanded leaves, (**b**–**d**) detail of necrotic patches on leaves.

**Figure 3 insects-13-00175-f003:**
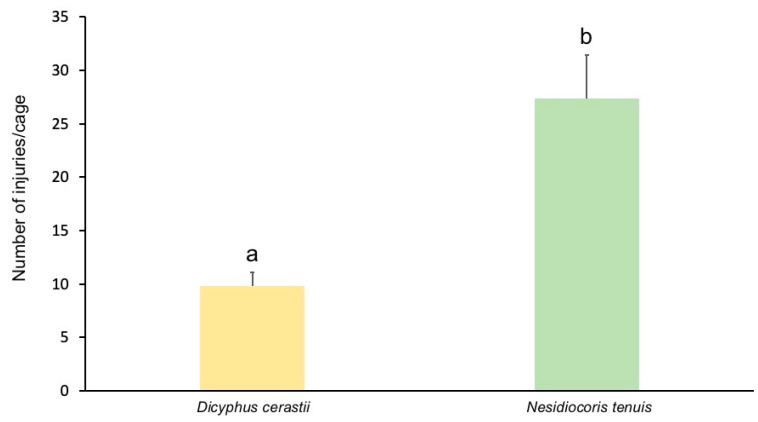
Number of plant injuries (necrotic rings or patches) per cage, by *Dicyphus cerastii* and *Nesidiocoris tenuis* on tomato plants (two plants/cage). Bars topped by different letters represent means that are significantly different (ANOVA, *p* < 0.05).

**Figure 4 insects-13-00175-f004:**
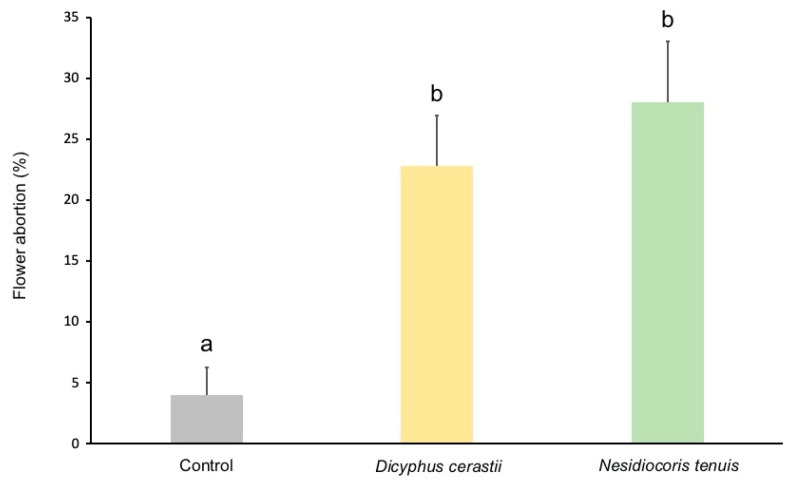
Percentage (+SE) of flower abortion (missing flowers/total number of flowers × 100) on control, *Dicyphus cerastii*, and *Nesidiocoris tenuis* tomato plants. Bars topped by different letters represent significantly different percentages (χ^2^, *p* < 0.05).

**Figure 5 insects-13-00175-f005:**
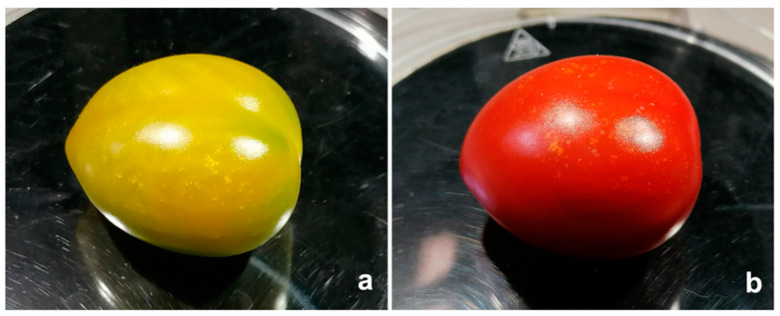
Feeding punctures in tomato fruit: (**a**) unripe fruit; (**b**) ripe fruit.

**Figure 6 insects-13-00175-f006:**
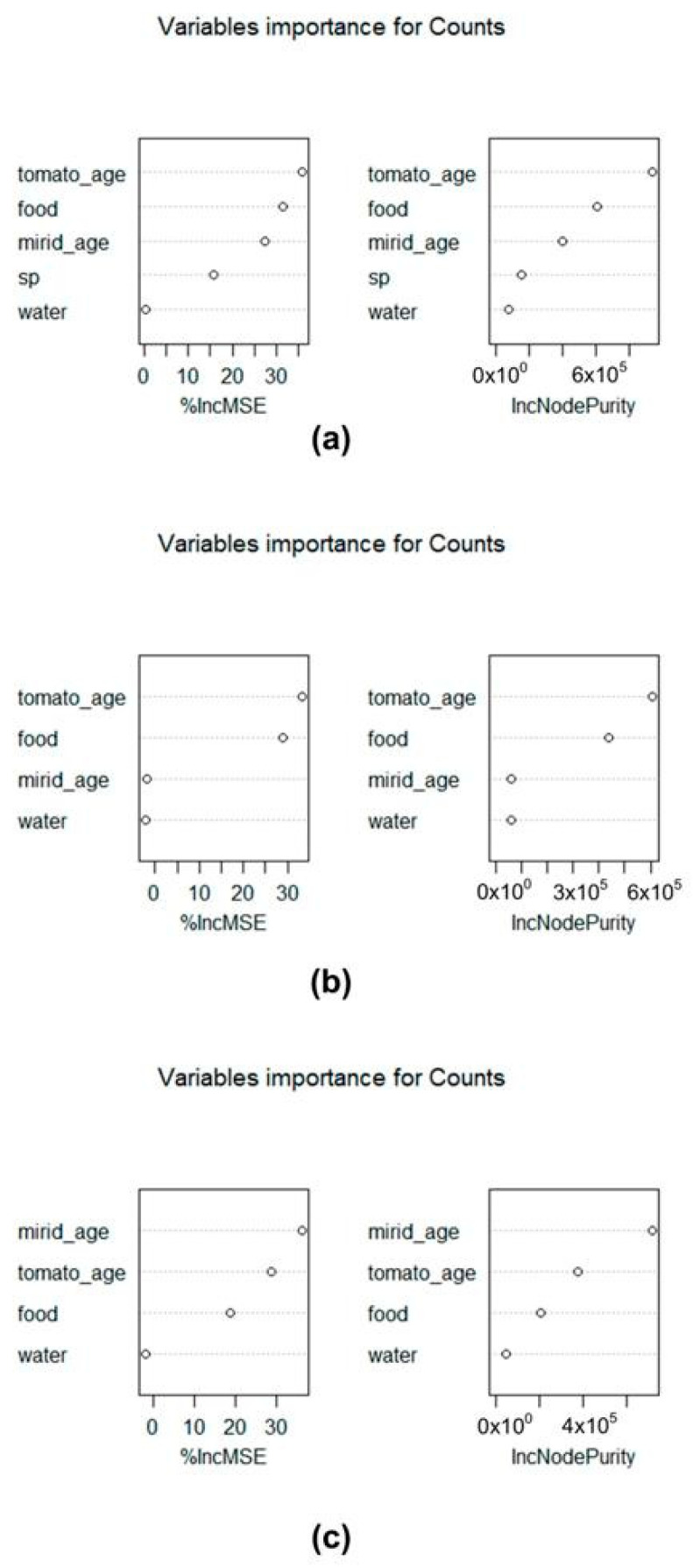
Variable importance plot from the random forest model (randomForest). The variables are ordered top-to-bottom as most-to-least important for an increase in feeding punctures (counts) on tomato fruit. (**a**) Using all datasets with data from both species, *Dicyphus cerastii* and *Nesidiocoris tenuis*; (**b**) database containing data collected only for *D. cerastii*; (**c**) database containing data collected only for *N. tenuis*. %incMSE: increase in mean square error of predictions as a result of the variable being permuted; inNodePurity: importance of each predictor variable.

**Figure 7 insects-13-00175-f007:**
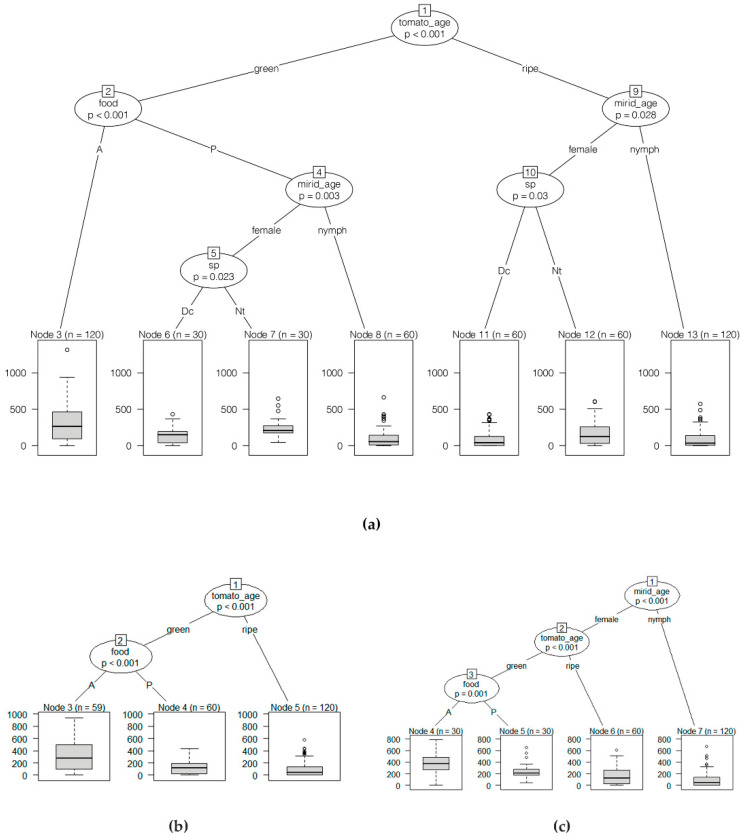
Classification trees from the conditional inference trees (ctree) model. For each internal node, input variable and P values are provided, the boxplot of the number of feed punctures is displayed for each end node. Numbers in boxes above the variable indicate the node number. Number above boxes (n) indicates number of fruit. (**a**) Using all datasets with data from both species, *Dicyphus cerastii* and *Nesidiocoris tenuis*; (**b**) database containing data collected only for *D. cerastii*; (**c**) database containing data collected only for *N. tenuis*. Dc: *Dicyphus cerastii*; Nt: *Nesidiocoris tenuis*; A: absence of food; P: presence of food.

**Figure 8 insects-13-00175-f008:**
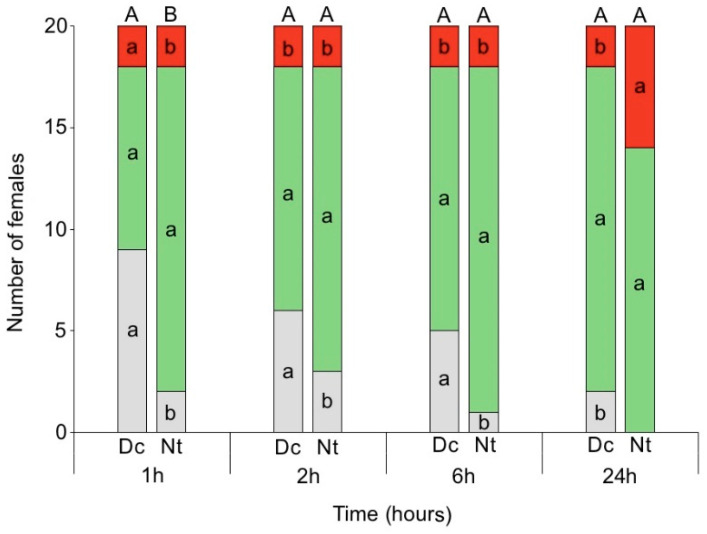
Number of *Dicyphus cerastii* (Dc) and *Nesidiocoris tenuis* (Nt) females on the young tomato plant (green), tomato fruit (red), or cup wall (grey) after 1 h, 2 h, 6 h, and 24 h. Bars topped by different letters for each observation time represent significant differences between species (Fisher’s exact test, *p* < 0.05); different letters within the same column indicate differences among location for each species (χ^2^, *p* < 0.05).

## Data Availability

Not applicable.

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
