# Peer review of "Half Friend, Half Enemy? Comparative Phytophagy between Two Dicyphini Species (Hemiptera: Miridae)"

_insects, 2022, doi:10.3390/insects13020175_

Round 1
Reviewer 1 Report
Dear authors, here are my comments:
1. Introduction:
- Paragraphs 5 and 6, those related with "Plant Damage" should be together.
- The last paragraph, it contains a clear objective, the comparison of two mirid bugs. However, you describe the three experiments, instead to explain the goal of each one. I recommend stating a start hypothesis such as: both mirid bugs behave similarly in absence of prey. After that you should describe your objectives in a general way such as: 1. Describe damages; 2. Damage factors; 3. Host location.
2. Material and Methods: (probably some answers will be clear, after check the results, however this section is important to reproduce your experiment, and should be clear enough).
-In the point 2.1, you explained that 4th and 5th instar nymphs were used. However, the mymphs were used only in experiment 2. Did you collect 4th and 5th instar nymphs, to individualize them until they molt to adults? Explain it, please!
-In experiment 1, your counting is per cage or per plant? I think it is not necessary to introduce food inside the cages, since you are looking for phytophagous damage. However, since food is present in all the treatments, it is okay, but is risky to find damages, since mirid bugs have preference for food.
-Fruit damage experiment: Explain better your 15 replications (5 N. tenuis + 5 D. cerastii + 5 Control?). You said at the beginning 3 adults o 3 nymphs, and then you said nymphs molted were discarded. Explain it, please. Is it insect stage a factor? Is it fruit ripeness a factor, too? Are 15 replications for each factor?
-In experiment 3, I understand that there is no food, just plant and green tomatoes. Isn't it?
-Data analysis is not clear in experiment 2. It seems that you check for ripeness preference, too. However, this is not clear according to the analysis. Explain it better.
3. Results:
-Did you check for ANOVA assumptions in point 3.1. Probably with a transformation you can get differences between both mirid bugs, since Pvalue it was very close to a significant value.
-Explain better, figure 6. What is %IncMSE and IncNodePurity?
-In FIgure 7, what is the menaing of "n"? number of fruits?
-In point 3.3, you need to put also the Fisher's exact test values for 2h, 6h and 24h.
4. Discussion:
-In the paragraph 12, you said that you observed feeding punctures on the tomato fruits. I did not see this on your results. I think it is important to know the feeding punctures on both plant and fruit. That data give you more information than punctual observations.
Author Response
Dear reviewer 1,
Thank you for dedicating part of your time to contribute to the improvement of our work. We are sure that your contribution was essential for the final quality of our work. All your suggestions have been considered and below I send our answers (in blue) to each of your questions.
Best regards,
Paula Souto
- Introduction:
- Paragraphs 5 and 6, those related with "Plant Damage" should be together.
R: Done.
- The last paragraph, it contains a clear objective, the comparison of two mirid bugs. However, you describe the three experiments, instead to explain the goal of each one. I recommend stating a start hypothesis such as: both mirid bugs behave similarly in absence of prey. After that you should describe your objectives in a general way such as: 1. Describe damages; 2. Damage factors; 3. Host location.
R: Done
- Material and Methods: (probably some answers will be clear, after check the results, however this section is important to reproduce your experiment, and should be clear enough).
-In the point 2.1, you explained that 4th and 5th instar nymphs were used. However, the nymphs were used only in experiment 2. Did you collect 4th and 5th instar nymphs, to individualize them until they molt to adults? Explain it, please!
R: We added “for all three bioassays” and “(see 2.3 Fruit damage bioassay)” on section 2.1.
-In experiment 1, your counting is per cage or per plant? I think it is not necessary to introduce food inside the cages, since you are looking for phytophagous damage. However, since food is present in all the treatments, it is okay, but is risky to find damages, since mirid bugs have preference for food.
R: We added “the total” and “for every cage”. We used food to simulate a more realistic scenario in protected crops and we previously observed that in the field damage can occur even in the presence of prey.
-Fruit damage experiment: Explain better your 15 replications (5 N. tenuis + 5 D. cerastii + 5 Control?). You said at the beginning 3 adults o 3 nymphs, and then you said nymphs molted were discarded. Explain it, please. Is it insect stage a factor? Is it fruit ripeness a factor, too? Are 15 replications for each factor?
R: The paragraph was restructured.
-In experiment 3, I understand that there is no food, just plant and green tomatoes. Isn't it?
R: Correct.
-Data analysis is not clear in experiment 2. It seems that you check for ripeness preference, too. However, this is not clear according to the analysis. Explain it better.
R: We hope that restructuring of the Fruit damage experiment paragraph made it clearer.
- Results:
-Did you check for ANOVA assumptions in point 3.1. Probably with a transformation you can get differences between both mirid bugs, since Pvalue it was very close to a significant value.
R: Yes. Data was normally distributed and homogeneous. No need to transform.
-Explain better, figure 6. What is %IncMSE and IncNodePurity?
R: We added the information in the legend of figure 6.
-In Figure 7, what is the meaning of "n"? number of fruits?
R: Number of fruits. Included in the legend of figure 7.
-In point 3.3, you need to put also the Fisher's exact test values for 2h, 6h and 24h.
R: In fact, the Fisher’s test was used for all the cases (1 h, 2 h, 6 h, and 24 h) because in all cases there were cells with expected count lower than 5.
- Discussion:
-In the paragraph 12, you said that you observed feeding punctures on the tomato fruits. I did not see this on your results. I think it is important to know the feeding punctures on both plant and fruit. That data give you more information than punctual observations.
R: The information was already in the second paragraph of section 3.3 of the results, but we restructured the sentence. Still, in this bioessay the objective was to understand if there was a difference between the species in terms of position and not in terms of damage.

Reviewer 2 Report
This study compares two zoophytophagous mirids N. tenuis and D. cerastii damages on tomato plants and fruits. This is well designed, well analysed and well written, I apreciated the reading of it.
I only have a few minor corrections to suggest :
-figure 6 : please make it more readible, in the same level than the rest of your figures
-tomato fruits used in your trials : in the text you say that you use fruits coming from "organic farming", could you please detail your fruit checking protocol, as I am sure that in an organic farming production there might be some insects or deseases in some fruits, and even early event of puncture in very young fruits before collection in case mirids were used (do you know which biological control agents were present in the production greenhouses ?). Did you isolate flowers before fruits production to insure clean fruits ?
in case you didnt isolate them early, please discuss this part in the discussion.
_random forest analysis : if i remember correctly, in case of random forest analysis you decide how many times you want to "play the model" in order to generate the statistical tree, so could you please indicate the implementation number you used.
i think there is a bug in the automatic numbering in the reference list
mirid rearing: I think that for more clarity (in regard with Dr Chinchilla Ramirez work), since those details aren't present in your [3] ref either, that you provide in sup data a table with the geographic origin of the populations that were collected to initiate/refresh rearings. It might illustrate better the point you raised about heterogeneity of behaviors that could be related with poplations genetic.
Author Response
Dear reviewer 2,
Thank you for dedicating part of your time to contribute to the improvement of our work. We are sure that your contribution was essential for the final quality of our work. All your suggestions have been considered and below I send our answers (in blue) to each of your questions.
Best regards,
Paula Souto
-figure 6 : please make it more readable, in the same level than the rest of your figures
R: In fact figure 6 was small. We've re-arranged the graphics in the figure and hope it looks better.
-tomato fruits used in your trials : in the text you say that you use fruits coming from "organic farming", could you please detail your fruit checking protocol, as I am sure that in an organic farming production there might be some insects or diseases in some fruits, and even early event of puncture in very young fruits before collection in case mirids were used (do you know which biological control agents were present in the production greenhouses ?). Did you isolate flowers before fruits production to insure clean fruits ?
in case you didnt isolate them early, please discuss this part in the discussion.
R: We restructured the section on materials & methods, and included further details on the protocol of fruit inspection for the experiments.
_random forest analysis : if i remember correctly, in case of random forest analysis you decide how many times you want to "play the model" in order to generate the statistical tree, so could you please indicate the implementation number you used.
R: We used 1001 trees (ntree=1001) in our analysis. We added this information to the material and methods.
i think there is a bug in the automatic numbering in the reference list
R: Done.
mirid rearing: I think that for more clarity (in regard with Dr Chinchilla Ramirez work), since those details aren't present in your [3] ref either, that you provide in sup data a table with the geographic origin of the populations that were collected to initiate/refresh rearings. It might illustrate better the point you raised about heterogeneity of behaviors that could be related with populations genetic.
R: We added the following information in the Discussion section, together with the Dr. Chinchilla Ramirez work citation: “As the large majority of the individuals used to initiate, and all the ones used to refresh the rearings come from nearby locations (less than 45 km of linear distance), it is likely that the geographic origin is not a key determining factor in the high variability in feeding puncture number.”.
